# Unraveling the Metabolic Hallmarks for the Optimization of Protein Intake in Pre-Dialysis Chronic Kidney Disease Patients

**DOI:** 10.3390/nu14061182

**Published:** 2022-03-11

**Authors:** Patricia Gonzalez, Pedro Lozano, Francisco Solano

**Affiliations:** 1Project Manager, Fresenius Kabi España, Sociedad Anonima Unipersonal, Marina 16-18, 08005 Barcelona, Spain; 2Department of Biochemistry and Molecular Biology “B” and Immunology, Faculty of Chemistry, Campus de Espinardo, University of Murcia, 30100 Murcia, Spain; plozanor@um.es; 3Department of Biochemistry and Molecular Biology “B” and Immunology, IMIB (Murcian Institute of Health Research), Faculty of Medicine, Campus de Espinardo, University of Murcia, 30100 Murcia, Spain

**Keywords:** CKD, enteral nutrition, protein ingestion, mTOR, Leu, oxidative stress, muscular proteostasis

## Abstract

The daily amount and quality of protein that should be administered by enteral nutrition in pre-dialysis chronic kidney disease (CKD) patients is a widely studied but still controversial issue. This is due to a compromise between the protein necessary to maintain muscular proteostasis avoiding sarcopenia, and the minimal amount required to prevent uremia and the accumulation of nitrogenous toxic substances in blood because of the renal function limitations. This review underlines some intracellular and extracellular features that should be considered to reconcile those two opposite factors. On one hand, the physiological conditions and usual side effects associated with CKD, mTOR and other proteins and nutrients involved in the regulation of protein synthesis in the muscular tissue are discussed. On the other hand, the main digestive features of the most common proteins used for enteral nutrition formulation (i.e., whey, casein and soy protein) are highlighted, due to the importance of supplying key amino acids to serum and tissues to maintain their concentration above the anabolic threshold needed for active protein synthesis, thereby minimizing the catabolic pathways leading to urea formation.

## 1. Introduction

In chronic degenerative diseases related to nitrogen metabolism and excretion, such as chronic kidney disease (CKD), the optimization of the amount of protein ingested in the diet is especially relevant. The occurrence of metabolic disorders and the nature of the amino acid mixture resulting from protein digestion, the general state of the patient and the severity of the CKD implies that proper protein intake should be carefully considered. CKD patients are usually classified into five stages according to eGFR and albuminuria, but for purposes of urea and other nitrogenous molecules, the five stages might be condensed into two large groups. Those in pre-dialysis with a conservative therapy, and those with advanced CKD that undergo periodical dialysis. In this context, the Kidney Disease Quality Initiative-National Kidney Foundation (KDOQI-NKF) guidelines for nutrition recommends a protein intake of 0.6 to 0.8 g/kg/day for pre-dialysis patients with CKD with an energy intake of 30 kcal/kg/day. On the other hand, the amount of protein recommended for advanced CKD patients is higher than 0.8 g/kg/day (up to 1.0–1.2) to compensate for the increased loss of protein and amino acids during dialysis [1,2]. The degree of albuminuria is also a key factor when adjusting the daily amount of protein.

Focusing on pre-dialysis CKD patients, the optimization in protein intake is necessary because of two conflicting factors. On the one hand, this disease requires a protein restriction in the diet to minimize general azotemia, which involves an excess of nitrogenous toxic molecules in the blood, mostly ammonium or urea. These metabolites originate as consequence of the protein tissue metabolism, including intestinal microbiota. On the other hand, a reduction in protein intake to minimize the contribution of the urea cycle could lead to a protein malnutrition risk with a consequent deficit in the supply of essential amino acids that are required for the replacement of body proteins [3]. This deficit would give rise to a loss in the protein synthesis capacity, which especially affects muscle tissue, producing lean muscle loss and sarcopenia [4,5], but also to other tissues, such as hepatic albumin synthesis. In that way, a deficit of protein ingestion puts these patients at a high risk of malnutrition [1].

Thus, it is assumed that one of the main goals for pre-dialysis CKD patients is the optimization of the amount and quality of protein intake for maintaining muscular proteostasis without the promotion of ureagenesis. However, the protein intake in CKD patients should account for a series of physiopathology factors usually related to CKD conditions. Aside from paying attention to the nitrogen balance between ingested and excreted nitrogen, the patients usually present altered metabolic situations, such as hormonal dysregulation, metabolic acidosis, insulin resistance, oxidative stress, chronic inflammation and hyperphosphatemia [6,7,8,9,10]. Thus, the increase of the oxidative stress levels is a prominent feature in CKD patients that must be considered.

On the other hand, the source of protein intake should also be considered beyond the daily necessities of essential amino acids, because of the sensor functions and the protein synthesis they control [11,12]. The problem is complex and controversial despite the huge number of studies available. This review aims to highlight some metabolic insights on the main factors to be considered for a rational design of protein intakes to preserve muscular proteostasis in pre-dialysis CKD patients under conservative therapy. The goal of this contribution is not the proposal and quantitation of the amount of protein that should be administered, because there are many recommendations from scientific societies and nutrition committees in addition to those mentioned formerly, many others [13,14,15,16,17] and excellent and updated reviews for complying that point [18,19].

## 2. CKD Side Effects and Their Usual Associated Conditions

Figure 1 displays the main metabolic disorders that can be associated with CKD patients. These patients show a significant decrease in the capacity of excretion of the nitrogenous toxic substance and hyperuremia. This is the main reason why a low protein-diet is recommended, and the preservation of muscular protein is compromised. In addition to that, clockwise, CKD patients used to have decreased levels of anabolic hormones but an increase in the secretion of several growth factors. These alterations increase the threshold of the nutrient sensing pathways such as the mTOR in the muscular tissue and simultaneously induce kidney hypertrophy and fibrosis damage in the glomerular and renal tissue, vasoconstriction, and other relevant effects.

Other disturbances in these tissues include lowering the activation and beneficial action of vitamin D [20,21], inactivation of the differentiation of satellite cells needed for myocyte replacement [22], existence of metabolic acidosis [23], insulin resistance [24] and increase in oxidative stress [6,7,8,9,10]. The appearance of higher amounts of ROS and RNS (reactive oxygen and nitrogen species) induce the expression of two transcription factors, FoxO and NFκB [25,26]. These factors determine the presence of metabolic proteolysis and the secretion of a set of inflammatory cytokines that worsen the pathological conditions and increase the difficulties in maintaining muscular proteostasis. Vascular endothelium is also severely affected by oxidative stress. Endothelin-1 and other factors provoke vasoconstriction, while nitric oxide cannot counteract such effects. Conversely, the formation of peroxynitrites and other RNS/ROS causes significant damage, leading to a partial lost in the muscular vascularization [8].

## 3. Muscular Atrophy and Sarcopenia in CKD Patients

The tissue demanding the highest amount of amino acids for protein synthesis is the muscular tissue. In healthy subjects, approximately one third of the total daily amino acids are used by the muscles. Therefore, the muscular tissue is the most affected tissue in case of a low-protein diet recommended in pre-dialysis CKD patients. Muscular proteostasis is the result of an equilibrium between protein synthesis and protein degradation. Figure 2 shows the main factors and interactions that regulate both opposite processes in the myocyte.

### 3.1. Protein Synthesis

Protein synthesis, as for other anabolic processes, is controlled and coordinated by the mTOR (mammalian Target Of Rapamycin) complex via a complex versatile and interconnected signal pathway network [27,28]. Briefly, two requirements are essential for mTOR activation; the first one concerns growth signals due to the binding of growth factors (mainly IGF-1, Insulin, but also GH) to its corresponding membrane receptor, and the second involves nutrients availability. Growth factors activate the PI3K/Akt signal pathway [29], one of the requirements for mTOR activation. In turn, this signal produces a translocation of GluT4 transporters to the membrane, increasing the transport of glucose inside the cell. With regard to amino acids, there are several transport mechanisms for the different families (not shown in Figure 2 for clarity). Although a well-balanced and proportional intracellular mixture is essential for protein synthesis, leucine is particularly important. Leucine acts as a trigger-sensor for the activation of the mTOR pathway because of its high affinity towards Sestrin2 [30]. Sestrins are stress-inducible metabolic proteins that protect organisms against various noxious stimuli, including DNA damage, oxidative stress, starvation, endoplasmic reticulum stress, and hypoxia. Furthermore, Sestrin-2 regulates the metabolism mainly by the activation of the key energy sensor AMP-dependent protein kinase (AMPK) and inhibition of the mammalian target of rapamycin complex 1 (mTORC1) [12]. Thus, Sestrin2 is usually linked to another protein, GATOR2, as a Sestrin2-GATOR2 complex that remains inactive the mTOR. The formation of the Leu–Sestrin2 complex promotes the release and subsequent conversion of GATOR2 to GATOR1, as an active effector of the mTOR pathway [31,32]. This displacement of the Sestrin2-GATOR2 so as to release GATOR1, and some other factors, allows for the translocation of mTOR to the periphery of the lysosomal membrane, and the full activation of the phosphorylation of key proteins for anabolic processes (mostly SREBP for lipid synthesis [33] and S6K1 plus 4E-BP1 for protein synthesis [34]).

### 3.2. Protein Degradation

Proteolysis of the myocyte proteins is stimulated by some transcription factors (FoxO and NFκB) related to oxidative stress, a usual condition in CKD as previously discussed (Figure 1). Extracellular and intracellular ROS generated by NOX membrane reactions or mitochondrial respiration induce: (i) the expression of atrogenic genes, mostly atrogin-1 and MuRF1 [35]. They act as ubiquitin ligases and increase the protein degradation rate via the ubiquitin-proteasome system [36,37], including (ii) the formation of Sestrin2, which is a member of the family of defense-proteins that protect cells against the ROS damage that is able to bind to GATOR2 and block protein synthesis by maintaining mTOR as an inactive form. In that way, oxidative stress accelerates protein degradation and inhibits protein synthesis, so that the equilibrium between both processes is broken, and the muscular proteostasis is lost. From an overall point of view, oxidative stress induces Sestrin2 expression, and this led to the necessity of higher levels of Leu for GATOR1 release and mTOR activation. In turn, Leu availability to the myocyte is diminished due to the partial loss of vascularization described in Section 2. Altogether, the anabolic threshold in the CKD patient is higher than in healthy people, as the higher levels of Sestrin2 plus the lower supply of Leu through the blood cannot trigger the same degree of mTOR activation.

## 4. On the Sources of Protein in Enteral Nutrition

Diet proteins are the source of amino acids after their digestion. According to that, the important point to consider is the amount and quality of the protein intakes, because they determine the supply of a balanced mixture of all amino acids needed for protein synthesis in all tissues, mostly in muscles. This mixture should not be an identical concentration for the twenty protein amino acids, but proportional to the amino acid composition of the main proteins synthesized for protein turnover, such as myosin and actin for the muscle tissue. These amino acids are the branched chain Ile, Val, Leu, Met and Lys [38], which are mostly essential amino acids. Moreover, the serum concentration of these amino acids would be higher at the anabolic threshold as long as possible to facilitate protein synthesis. On the other hand, the appearance of limiting amino acids mean that the excess of other amino acids led them to be catabolic pathways with the subsequent activation of the urea cycle [36,39]. The catabolic pathway of amino acids should be minimized in CKD patients to decrease as much as possible uremia and the appearance of toxic nitrogenous compounds that are not easily excreted.

Accordingly, the nutritional quality of the ingested proteins suggests them to be the best for use, mostly in enteral nutrition. This nutritional quality is determined by several factors, mostly the protein digestibility-corrected amino acid score, which is related to the digestion rate and the content in the first limiting amino acid [13,40,41].

The usual sources of protein in enteral nutrition are whole milk, whey, casein and soy. Concerning the last one, some low-protein diets using soy and other vegetal proteins have been recently proposed as suitable alternatives to animal proteins in CKD patients [18]. However, there are some doubts in relation to the protein digestibility-corrected amino acid score. Those animal and vegetal proteins have different isoelectric points (pI), and this parameter is important for the digestion rate. For a protein, the pI value corresponds to a pH where this protein has usually minimal solubility and the aggregation/precipitation is facilitated. Therefore, the relationship between the pI and the pH in the digestive tract is an important parameter that greatly affects the solubility. This is directly related with the suitability as substrates for enzymatic hydrolysis, which determine the digestion time and post-prandial absorption of the released amino acids. This phenomenon is most important at the stomach for casein due to its acidic pI and the acidity of the gastric juice [42].

Dietary protein quality depends on the degree and rate at which such proteins are digested, absorbed as amino acids, and used in the tissues for new protein synthesis. According to this, the solubility degree of each protein in the acidic gastric juice determines its hydrolysis rate at the stomach, the transfer to the gut for complete hydrolysis and amino acid absorption.

Whey protein is very soluble to the gastric pH. Its transit to the duodenum is fast, as well as its hydrolysis rate, so that the amino acid bioavailability is high [43]. The concentration of those units in the blood increases rapidly but it is not long lasting. The anabolic threshold is reached easily, but it also decays quickly [44,45]. On the other hand, casein is insoluble at the acidic gastric pHs, and it tends to coagulate. Its hydrolysis is rather slow, and the increase in the concentration of serum amino acids is not sharp, but it is maintained for longer times [46]. This might induce a net protein synthesis for longer times, as long as the serum concentrations are higher than the anabolic threshold.

In that way, it has been proposed that the protein synthesis in muscle can be improved by the synergic action of whey and casein mixtures [47]. Enteral nutrition based in casein-whey mixtures would allow for a complementary and even synergist effect in the enhancement of the body protein synthesis. The digestion of those mixtures supplies serum concentration levels above the anabolic thresholds for longer periods. This is especially important in CKD patients that show higher thresholds for net protein synthesis in muscular tissue and a low amount of protein intake due to the renal failure.

On the other hand, soy proteins have a mean pI intermediate between the pIs values of casein and whey, so that its low solubilization at the gastric pH is not as high as casein and the hydrolysis and subsequent release of amino acids for its bioavailability is intermediate among the milk proteins. Alternatively, according to the amino acid composition, soy protein has a lower biological value than milk proteins. Compared with casein [48], soy protein is deficient in the essential amino acids Met and Lys and contains less branched chain amino acids; such differences are even greater in comparison to whey. In general, the biological value of vegetal proteins is lower than that of collagen from animals due to the low content of essential amino acids such as Leu, Lys and Met [49,50,51,52].

Despite those features, the use of soy protein in enteral nutrition is also frequently used as the main vegetal source of protein. Some studies comparing the post-prandial metabolism of those proteins in animal and also in humans highlight that the quality of soy protein is lower to that of casein and whey in healthy subjects [13,46,48], and the problems are even more pronounced when applied to CKD patients. Thus, the results clearly show that the urea production two hours after the ingestion is higher in the group fed with soy protein. A more detailed study measured whole-body protein kinetics, urea production and splanchnic Leu content after enteral intakes of isonitrogenous protein-based meals containing casein or soy protein [48]. Firstly, both intakes altered the protein metabolism from net protein breakdown to net protein synthesis, and consequently urea synthesis rates decreased during the consumption of both enteral meals. More interestingly, net protein synthesis was greater in the casein group than in the soy group, but the urea decrease was more significant in the group that consumed casein. Secondly, splanchnic Leu was also higher in the subjects that consumed casein than those that consumed soy protein. In conclusion, nitrogen of the soy protein is degraded to the urea at a greater proportion than casein protein, whereas the nitrogen of casein is incorporated into the body protein by a greater stimulation of protein synthesis.

The total content of phosphate (free and bound) is an important point to be considered in the diet or enteral supplements that contain casein, casein/whey mixtures or just milk because of several problems including osteopathy, abnormal serum data, and vascular calcification, collectively called CKD-mineral bone disease (CKD-MBD), which may be involved. [50,52,53]. Phosphate plays a critical role in bone formation, acid-base balance, and energy production, and its homeostasis is achieved by excreting the excess phosphate in the urine. However, the decrease in renal function that occurs for CKD patients prevents the maintenance of this homeostasis, with it being necessary to rigorously control phosphate intake [2]. To avoid related problems with kidney malfunction, the 2020 KDOQI-NKF guidelines recommended an intake of phosphate that keeps serum phosphate levels within normal ranges (3.4–4.5 mg/dL), paying special attention to the dietary phosphate for the case of hyperphosphatemia [2,17]. As the bioavailability of phosphate depends on dietary sources (i.e., 20–40% for plant foods; 40–60% for animal protein; and 100% for phosphate found in additives and processed foods), choosing phosphate-containing foods lower in phosphate bioavailability is recommended [2]. 

## 5. Conclusions

Nutritional strategies in CKD patients requires flexible planning to look for the optimal ingestion of protein and reconcile the minimal muscular necessities to avoid loss of lean muscle mass and simultaneously to ameliorate hyperuremia and nitrogenous toxic compounds due to limitations of the renal function. Physiological conditions or the grade of severity of CKD should be taken into account. As much as those conditions are worse, the sensitivity of detection and the usage of amino acids for muscular protein synthesis is lower. The inherent oxidative stress associated with CKD should be minimized in order to achieve minimal cellular damage. ROS and pro-inflammatory cytokines induce the expression of Sestrin2, and activation of muscular proteolysis mediated by the ubiquitin/proteasome system. The administration of essential amino acids, especially Leucine, should be provided to block Sestrin2 action, thereby allowing for the activation of mTOR and muscular protein synthesis.

Quality of protein (determined by source, digestibility and content and ratio of amino acids) affects nutrient sensing pathways such as the mTOR. Aside from the amount, the source of the ingested protein is important due to the different digestion features and composition of amino acids, which greatly affect the metabolic fate of those units. They could be used as direct units for muscular protein synthesis or catabolized in liver and kidney to increase urea formation. The serum concentration of amino acids in the postprandial period should be balanced and above the threshold compatible with anabolic processes, that can be higher in CKD patients in comparison to healthy subjects. Essential amino acids, such as Leu or Lys, are particularly important. Leu is needed for the adequate activation of mTOR in the presence of oxidative stress, and consequently Sestrin2. On the other hand, an excess of one amino acid would lead it and even others to their respective catabolism pathways. Glucose or ketone bodies would be formed from the carbon chains, but the nitrogen would be fated to the urea which would worsen the situation of CKD patients. According to this, whey, casein, and soy are used for CKD patients as sources of amino acids in enteral diets, in that order of efficiency.

## Figures and Tables

**Figure 1 nutrients-14-01182-f001:**
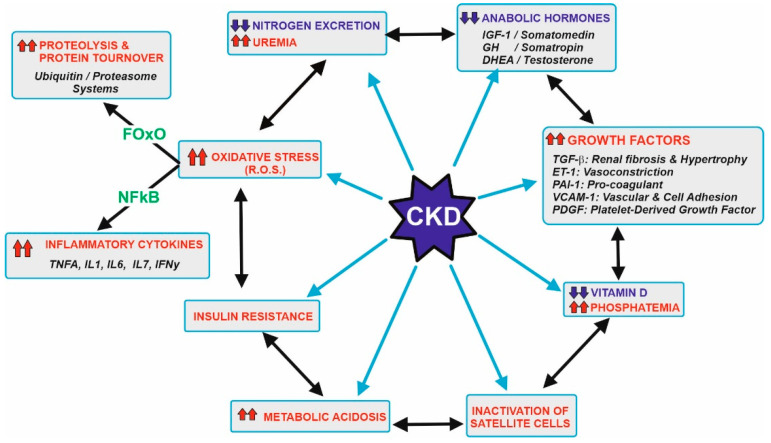
Metabolic abnormalities usually associated with patients with CKD. Starting at the left upper side, the gradual loss in kidney function leads to a decrease in the efficiency of nitrogen excretion and subsequent uremia. This should be related to recommend a control in the diet protein ingestion, which is the main goal of this review. Other abnormalities include a decrease in the levels of anabolic hormones, increase in a series of growth factors, decrease vitamin D and its effects, inactivation of muscle satellite cells needed for appropriate myocyte turnover, metabolic acidosis, and the development of insulin resistance, hyperglycemia and oxidative stress. Most of these effects are also interconnected by a complex network of signals. Oxidative stress induces transcription factors (majorly FOxO and NFkB) that increase damage in several tissues, including muscles due to higher proteolysis and the secretion of inflammatory cytokines. Further details are beyond the scope of this review, but appropriate references are cited at the text. Upward arrows in red indicate an increase, while downward arrows in blue indicate a decrease.

**Figure 2 nutrients-14-01182-f002:**
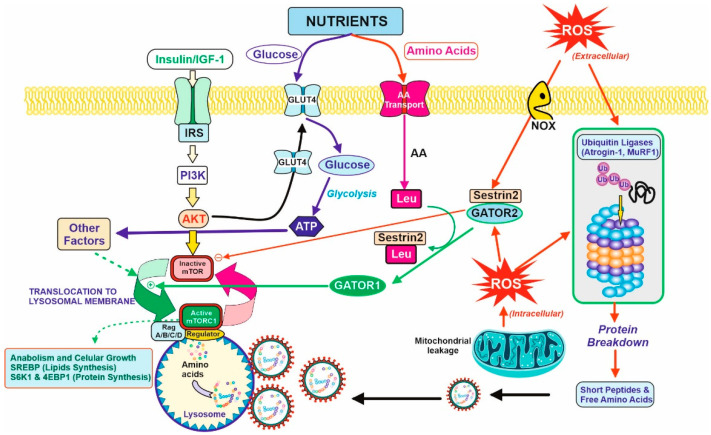
Balance between anabolic and catabolic signals in the myocyte. Nutrients and growth factors induce mTOR activation for protein synthesis. However, oxidative stress is caused by extracellular or intracellular ROS and accelerated protein degradation by the ubiquitin–proteasome system (by the induction of atrogenic genes. They are ubiquitin ligases (Atrogin-1 and MuRF1) that induce the formation of the protective protein Sestrin2 and diminish protein synthesis (right figure). Nutrients supply glucose for energy production (ATP) and amino acids for protein synthesis or catabolism, producing further energy and nitrogenous toxic compounds, mostly urea. Due to the last process, CKD patients should have limitations in terms of protein intake, and therefore, limits the bioavailability of amino acids. In this regard, Leu is partially important. Leu is needed for the activation of mTOR activity in the presence of Sestrin2 protein due to the oxidative stress situation. The binding of Leu to Sestrin2 promotes the action of GATOR1 for mTOR activation, although translocation to lysosomal membrane and other factors are required for complete activation and triggering anabolic processes (lipids and protein synthesis, left on the figure). CKD patients should have a counterbalance between the degradative and anabolic signals in the myocyte to avoid lean muscle loss with the minimal amount of protein to avoid excessive urea formation.

## Data Availability

Not applicable.

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
