# Peer review of "Unraveling the Metabolic Hallmarks for the Optimization of Protein Intake in Pre-Dialysis Chronic Kidney Disease Patients"

_nutrients, 2022, doi:10.3390/nu14061182_

Round 1

Reviewer 1 Report

This minireview, while presenting well-written parts, seems to send some misleading messages to the readers. In fact, the authors describe chronic kidney disease patients and they state that these patients must have a reduced protein intake without making a distinction between patients under conservative therapy and in renal replacement therapy. This distinction is fundamental as, according to the guidelines, if a reduced protein intake is recommended in patients under conservative therapy, conversely in renal replacement therapy patients the protein intake must be increased.

Furthermore, even the concepts relating to the intake of phosphorus are not properly correct (see lines 238-242). Therefore, it would be advisable for the authors to correctly argue these concepts, distinguishing between patients in renal replacement therapy and under conservative therapy

Minor comments:

Enter the list of acronyms

Line 29 correct with “ nitrogen metabolism and excretion”

Line 35 replace with synonym reduce

Line 41 correct high nutrition risk with “ high malnutrition risk”

Line 48 insert “metabolic” before acidosis

Lines 53-55 Rephrase

Line 60  correct “ including hemodialysis..” with “ Patients in renal replacement therapies ( like hemodialysis or peritoneal dialysis).

Line 83 Replace Increase of ..with “ the presence of metabolic…”

Line 98 Delete the first “That”

Line 157 correct “after its.. “ with “ after their…”

Line 159 correct “it” with..“ they”

Line 238 correct “guides” with “guidelines”

Line 262 correct “healthy human “ with “healthy subjects”

Author Response

Reviewer 1.

  1. This minireview, while presenting well-written parts, seems to send some misleading messages to the readers. In fact, the authors describe chronic kidney disease patients and they state that these patients must have a reduced protein intake without making a distinction between patients under conservative therapy and in renal replacement therapy. This distinction is fundamental as, according to the guidelines, if a reduced protein intake is recommended in patients under conservative therapy, conversely in renal replacement therapy patients the protein intake must be increased.

First of all, the reviewer is right. We thank to the reviewer for the thoughtful comment related the different protein requirement between patients under conservative therapy and in renal replacement therapy, and we totally agree with the necessity to distinguish the type of CKD patients. We did it at the revised version. This can be seen in the title, abstract (line 13), introduction (lines 31-44) the main aim of the mini-review (lines 65-70) and several other sections throughout the manuscript. Our work is essentially focused on pre-dialysis CKD patients under conservative therapy. We understand that this point was not clearly stated in the first version, as the reviewer remarked. Consequently, the text has been refocused and modified, and two new references (1 and 2) have been introduced. The situation of patients in dialysis and renal replacement therapy is different in circumstances, therapy and protein intake. That situation requires a deeper and systematic study of others interactive biochemical and physiological parameters related to the nitrogenous molecules lost during the dialysis.

  1. Furthermore, even the concepts relating to the intake of phosphorus are not properly correct (see lines 238-242). Therefore, it would be advisable for the authors to correctly argue these concepts, distinguishing between patients in renal replacement therapy and under conservative therapy

In agreement with reviewer suggestions, concepts related with the intake of phosphate have been properly modified, being focused for pre-dialysis CKD patients under conservative therapy (see lines 255-269). A new reference (53) has been added to clear up the specific stages of CKD patients we are referring to.

  1. Minor comments: Enter the list of acronyms: A list of acronyms can be found at the end of the manuscript (see lines 301-312)

Line 29 correct with “ nitrogen metabolism and excretion”. Done it

Line 35 replace with synonym reduce.  Done it

Line 41 correct high nutrition risk with “ high malnutrition risk”. Done it

Line 48 insert “metabolic” before acidosis. Done it

Lines 53-55 Rephrase . Done it

Line 60, correct “ including hemodialysis..” with “ Patients in renal replacement therapies ( like hemodialysis or peritoneal dialysis). This part has been eliminated due to the re-focusing of the minireview.

Line 83 Replace Increase of ..with “ the presence of metabolic. Done it

Line 98 Delete the first “That” Done.

Line 157 correct “after its.. “ with “ after their. Done it

Line 159 correct “it” with..“ they” Done it

Line 238 correct “guides” with “guidelines” Done it

Line 262 correct “healthy human “ with “healthy subjects” .Done it

In turn, some other hypos and grammatical errors detected by an English native speaker have been corrected. We regret the inconveniences.

Reviewer 2 Report

Authors reviewed and discussed the suitable quality of protein in patients with CKD. The review is of some merit but the current version needs several revisions.

Major points

  • This review discussed about patients with CKD but it does not specify the degree of CKD. Do authors focus on patients with dialysis or non-dialysis CKD, if the latter, what stages? Since the patients with mild or advanced non-dialysis CKD or dialysis CKD do not have similar protein balance.
  • The authors discuss mainly about the quality of protein, but do not discuss how the doctors prescribe the diet/enteral supplements or how the patients select the foods available to them. This is very important since even if we know the casein/whey mixtures is superior, it is useless if these mixture is readily available in foods sold in market or is available as enteral preparations (although many patients cannot afford them). Authors at least how they can get high quality proteins in foods or enteral preparations which is readily available.
  • Also, authors should discuss on the quantity of the protein they should intake.

Minor points

  • CKD (chronic kidney disease) seemed to be miss spelled as CDK in several occasions, which needs to be corrected.
  • English is extensively reviewed by native English speaker.

Author Response

Reviewer 2.

  1. Authors reviewed and discussed the suitable quality of protein in patients with CKD. The review is of some merit but the current version needs several revisions.

We thank reviewer for kind comments with regards the interest of our contribution.

  1. This review discussed about patients with CKD but it does not specify the degree of CKD. Do authors focus on patients with dialysis or non-dialysis CKD, if the latter, what stages? Since the patients with mild or advanced non-dialysis CKD or dialysis CKD do not have similar protein balance.

This point was risen by both reviewers. We have the same answer. First of all, the reviewer is right. We thank to the reviewer for the thoughtful comment related the different protein requirement between patients under conservative therapy and in renal replacement therapy, and we totally agree with the necessity to distinguish the type of CKD patients. We did it at the revised version. This can be seen in the title, abstract (line 13), introduction (lines 31-44) the main aim of the mini-review (lines 65-70) and several other sections throughout the manuscript. Our work is essentially focused on pre-dialysis CKD patients under conservative therapy. We understand that this point was not clearly stated in the first version, as the reviewer remarked. Consequently, the text has been refocused and modified, and some new references have been introduced. The situation of patients in dialysis and renal replacement therapy is different in circumstances, therapy and protein intake. That situation requires a deeper and systematic study of others interactive biochemical and physiological parameters related to the nitrogenous molecules lost during the dialysis.

  1. The authors discuss mainly about the quality of protein, but do not discuss how the doctors prescribe the diet/enteral supplements or how the patients select the foods available to them. This is very important since even if we know the casein/whey mixtures is superior, it is useless if these mixture is readily available in foods sold in market or is available as enteral preparations (although many patients cannot afford them). Authors at least how they can get high quality proteins in foods or enteral preparations which is readily available. Also, authors should discuss on the quantity of the protein they should intake.

As suggested by reviewer, a better discussion about the amount and quality of protein intake for pre-dialysis CKD patients under conservative therapy has been included (see lines 31-44, and lines 65-70) as well as two new references (1 and 2). However, it should be noted that we consider out of the scope of this work how the doctors would prescribe composition and amount of diet/enteral supplements to patients or how the patients select the foods available to them. We assume that we are not nutritionists or nephrologists, and this is not our area of expertise. There are many reviews, books and guidelines of nutrition and medical societies to satisfy these criteria. We provide several references to cover this aspect (refs.3,4,13,14,15,16,17,50, 54). It is difficult we can improve that guidelines. This work is essentially focused for understanding the key metabolic parameters that control protein synthesis under reduced protein intake therapy as results of kidney malfunction at initial stages of CKD. Oxidative stress, the role of leucine as sensor amino acid for regulation of muscular proteostasis and the protein score related with the amino acid content and digestibility of the protein are the main parameters that we have discussed in this contribution, rather than a practical nutritional advice. At this point, a better explanation on the important role of sestrins in this process has also been added (see lines 130-134, as well as two new references 11 and 12). It is obvious that we did not establish our point of view. We regret that, and we hope it should be clear after your report.

  1. Minor points. CKD (chronic kidney disease) seemed to be miss spelled as CDK in several occasions, which needs to be corrected.

We are sorry for the misspelling in the chronic kidney disease acronym. All CDK have been corrected to CKD

  1. English is extensively reviewed by native English speaker.

The manuscript has been read and corrected by a native English speaker. 

Round 2

Reviewer 2 Report

Revised well enough to the comments. No further suggenstions.